# Machine Learning Project Proposal

**Teh Kai Jun** *
2024389001

**Shei Pern Chua** *
2024280203

**Thai Zhen Leng** *
2024280033

## 1 Background

Large Language Models (LLMs) have become essential tools across diverse sectors such as healthcare [1], finance [2] and education [3] due to their remarkable achievements. They had demonstrated high adaptability and efficiency for improving workforce's efficiency from text generation, complex reasoning, few-shot learning, protein modeling and etc [4, 5]. A study reported that ChatGPT had reached 600 millions of user visits monthly, which indicates the society's reliance on large language model [6].

Due to the high integration of LLMs across most industries, data safety and privacy concerns arise [7]. Malicious attackers exploit LLMs to execute harmful activities such as accessing confidential company data [8] or generating harmful contents [9]. Therefore, many models undergo rigorous alignment processes to mitigate these risks prior to public deployment [10].

Unfortunately, these commercial LLMs still remain susceptible to several attacks [11, 12, 13]. Wei, Haghtalab, and Steinhardt [14] mentioned restricted behaviors as actions that an safety aligned LLM has to avoid, such as providing harmful information which act as a threat to the society. However, attackers are able to exploit the models' vulnerabilities by crafting specific prompts to bypass these safety restrictions, leading to what's known as a **jailbreak attack**.

With the goal to discover the vulnerabilities of large language models (LLMs) towards jailbreak attacks, this project aims to provide insight towards strengthening LLM defenses by employing several advanced jailbreaking techniques. Given the growing integration of LLMs across industries, it is crucial to ensure the models' robustness in maintaining aligned and safe behaviors, since ChatGPT has a huge impact and adoption in various business applications [15]. Therefore, we selected the jailbreaking track from CLAS 2024: The LLM and Agent Safety Competition at NeurIPS 2024.

## 2 Definition

### 2.1 Mathematical definition

In a jailbreaking attack, the goal is to modify an initial prompt $P$ into a new prompt $P'$ to elicit restricted or unauthorized responses from a model $L$. For a successful attack, the modified prompt $P'$ must satisfy the condition $L(P') \approx O_{res}$, where $O_{res}$ represents a target restricted response, while $L(P) \neq O_{res}$ in most conditions.

### 2.2 White-box and Black-box

In white-box attacks, where the model weights $\theta$ are visible to attackers, the prompt modification can be framed as an optimization problem. In black-box attacks, attackers do not have access to the model weights $\theta$ and training data, limiting them to observe the output $L(P)$ for different inputs.

### 2.3 Targeted Models

The CLAS-2024 jailbreak competition [16] provides an open-source safety-aligned model (Llama-3 8B) as well as a closed-source aligned model for evaluation. Our approach expands the model set to include additional models such as GLM [17], Qwen [18], Mistral [19], along with others to broaden the evaluation scope.

---

*Equal contribution.

Preprint. Under review.

## 2.4 Evaluation Metrics

There are several constraints for the prompt to prevent cheating. 1) All tokens from the original prompt must be present in the submitted prompt and follow the original order to prevent potential cheating. 2) The perplexity change should not exceed 100. 3) A maximum of 20 tokens may be added.

The harmfulness score $J(L)$ of each submitted prompt will be evaluated using the HEX-Phi scoring system [20], ranging from 1 to 5. Prompts being rejected will get a zero score. To compute the stealthiness score $S(L)$, each jailbreak prompt and its original prompt are converted into their TF-IDF (Term Frequency–Inverse Document Frequency) features. The stealthiness score $S(L)$ is the cosine similarity between the TF-IDF features for each pair of jailbreak prompt and its original prompt and then take the average over all these pairs. The score for model L is $0.84 \times J(L) + 0.16 \times S(L)$.

## 3 Related Work

### 3.1 White-box Attack Techniques

This section focuses on discussing several latest white-box attacks for LLMs. This method allows attackers to optimize effective adversarial prompts by using model's known parameters such as logits or gradient-based optimization to fine-tune the prompts.

Zhou et al. [21] proposed Greedy Coordinate Gradient (GCG), an effective gradient-based jailbreak attack for aligned LLMs. This methods appends adversarial suffixes to prompts by iteratively replacing tokens with optimal replacements, which maximize the likelihood of harmful responses based on gradient evaluations. As a result, these prompts are often unreadable, which have higher chances to be rejected by model defenses targeting high perplexity inputs.

Therefore, several methods [22, 23] had been proposed that focuses on generating semantically meaningful adversarial prompts. These prompts not only increased the stealthiness of the attacks, but it also improves the chances of jailbreak. Although these methods are effective in jailbreaking, they fully rely on the internal model information, which is not applicable in commercial LLMs.

### 3.2 Black-box Attack Techniques

In black-box attacks, attackers design adversarial prompts to manipulate model in making incorrect predictions without any access towards models' parameter or training data. One of the technique is Scenario Nesting, which embeds malicious prompts within sentences with benign contexts [9]. For example, DeepInception [24] uses LLM's personification to bypass safety protocols by embedding prompts in a hypnotic scenario that generates harmful responses. Furthermore, ReNeLLM [25] rewrites harmful prompts and embed them in code completion or text continuation tasks, which further lead the model to generate the remaining malicious content.

Another technique commonly used in black-box attacks is LLM-based generation, where multiple models collaborate to create adversarial prompts. Several studies [26, 27] utilize this concept by focusing on automating adversarial prompt generation using fine-tuned LLMs to bypass security measures.

## 4 Proposed method

For training and testing data, we'll use the 100 harmful prompts provided by CLAS-2024 [16] which contains categories such as hate speech, privacy violations and etc. These prompts will be rejected by the model due to potential harmful outputs. We'll benchmark our approach with several existing white and black-box adversarial attack techniques not limited to techniques mentioned in Section 3 to establish a baseline for this research project.

Based on the advancement of existing work, we propose two main fusion LLM-based attack methods, fine-tuning method and accuracy selective method. Firstly, we will replicate state-of-the-art LLM-based generative attack methods to automate adversarial prompts generation, which serves as datasets. Furthermore, we select a group of victim models as targets to simulate a black-box attack environment and identify an attacker model of comparable size to these victim models.

Given a set of victim models $\{V_1, V_2, ...\}$, we choose an attacker model $A$ of similar size to the victim models. We aim to fine-tune model $A$ to effectively generate/modify harmful prompts into adversarial ones by selecting and integrating various prompt injection techniques. We will identify the most effective prompt injection method for each category of harmful content and use these as our training dataset.

Our alternative approach involves in evaluating the attack accuracy of each attacks on $\{V_1, V_2, ...\}$ using the metric mentioned on Section 2.4. Specifically, we measure the attack accuracy of each prompt. The goal is to identify the top 100 highest-accuracy prompt based on the consistent success across all victim models. These selected prompts will then be deployed against the secret model. This approach maximizes the likelihood of bypassing its alignment defenses.

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
