# OpenReview forum: "CLAS-2024: The Evolution and Tactics of Jailbreaking Attacks"
_tsinghua.edu.cn/THU/2024/Fall/AML — THU 2024 Fall AML Submission_

### Official Review · ~Guilherme_Félix_Diogo1 · 2024-11-06
**Clear explanation of the proposal**

**Rating:** 9
**Confidence:** 4

**Review:**

This proposal presents a persuasive argument on a critical aspect of the security of LLM, that is, jailbreak attacks. In terms of structure and presentation, the project is well structured and offered with an extensive explanation on white-box and black-box attack methodologies, which is quite commendable. A favourable aspect of the metrics of evaluation during the research is the HEX-Phi scoring system, which allows for definite results to be achieved. Overall, the proposal is fit in terms of technology and current market demand.

---

### Official Review · ~Aleksandr_Algazinov1 · 2024-11-06
**Clear, promising, and relevant**

**Rating:** 10
**Confidence:** 4

**Review:**

The proposal is well-written and easy to follow. The problem is relevant and related to widely-used LLMs. This makes the work potentially useful if done successfully. The methodology is clearly explained, and there is no doubt that authors have a clear plan to achieve their goals.

---

### Official Review · ~Ethan_Wei_Yuxin1 · 2024-11-08
**Strengthening Large Language Models: A Comprehensive Review of Jailbreak Attack Defenses**

**Rating:** 8
**Confidence:** 3

**Review:**

The paper does a solid job addressing most of the core requirements for our machine learning course project. It focuses on a real-world issue—strengthening the defenses of large language models (LLMs) against jailbreak attacks—and provides clear sections covering the background, mathematical definitions, related work, methodology, and evaluation metrics. That said, there are a few areas where it could align even better with the course guidelines. For instance, expanding on the motivation for the chosen techniques would make the approach more compelling, and specifying datasets or detailing baseline methods could strengthen the proposed method section. Finally, it would be helpful to clarify the broader impact of improving LLM defenses, especially given their increasing use across industries. Making these adjustments would bring the proposal in closer alignment with what’s expected for the assignment.

---

### Official Review · ~Jia-Nuo_Liew1 · 2024-11-08
**Comprehensive Proposal**

**Rating:** 10
**Confidence:** 5

**Review:**

The proposal provides a comprehensive overview of the importance of LLM safety.

Background: Provided a clear and concise thorough overview of the increasing reliance on LLMs in various industries and highlighted the risk of privacy and data safety. It effectively establishes the motivation for the study and links it to real-world concerns.
Definition: Provided clear mathematical and conceptual framework. The explanation of white-box vs black-box attack scenarios is useful.
Related Works: Provided a structured view of the current jailbreak methods. Describe how the current method can exploit model vulnerabilities, and justify the project.
Proposed Method: Outlined a clear approach for training and evaluating adversarial prompts using both white-box and black-box settings.

---

### Official Review · ~Kehan_Zheng1 · 2024-11-11
**Good Proposal**

**Rating:** 9
**Confidence:** 4

**Review:**

This proposal addresses an important issue in the field of LLMs by focusing on the vulnerabilities that allow jailbreaking attacks. The structured methodology, leveraging both white-box and black-box adversarial techniques, provides a solid framework for identifying and evaluating model weaknesses, which is both practical and innovative. It is better that the project focuses on not only the attack but also the defense against such attacks, and take some SOTA models like o1 into your test models.

---

### Official Review · ~Ziyad_Fawzy1 · 2024-11-11
**CLAS-2024: The Evolution and Tactics of Jailbreaking Attacks**

**Rating:** 10
**Confidence:** 5

**Review:**

The authors plan to advance model robustness against harmful and undesired content by advancing attack techniques.

This authors highlight a crucial and pressing topic: LLM safety. The authors' arguments are rigorous and easy to follow. They have also conducted a detailed and comprehensive literature review. This paper's outlook is promising, and its margins of contribution are wide.

---

### Official Review · ~Yida_Lu1 · 2024-11-11
**A good attempt on jailbreak attack**

**Rating:** 9
**Confidence:** 4

**Review:**

This study attempts to conduct a jailbreak attack research on several models with existing black-box and white-box methods and fine-tune an LLM attacker to automate the jailbreak process. The proposal is well-structured and is a good attempt to analyze the effectiveness of different attack methods, and an automated attacker will definitely facilitate the jailbreaking of LLMs. On the other hand, I'm curious about whether the attacker can learn from the optimization-based methods such as GCG and how to ensure the attack success rate of the attacker since some attack methods are sensitive to the template they use.

---

### Official Review · ~Liu_Yiyang1 · 2024-11-11
**Well defined, detailed and clear proposal**

**Rating:** 9
**Confidence:** 4

**Review:**

The proposal demonstrates an innovative approach to discovering LLM vulnerabilities by exploring both white-box and black-box attack methodologies. The fusion approach harbors great potential to achieve unprecedented accuracy and stealth. The proposal offers a detailed mathematical definition of jailbreak attacks, distinguishing between white-box and black-box scenarios. It also outlines the HEX-Phi scoring system and stealthiness criteria, providing a well-defined evaluation framework. All these factors combined makes for a promising project!

---

### Official Review · ~Qihang_Cen1 · 2024-11-12
**Innovative work and good proposal**

**Rating:** 10
**Confidence:** 4

**Review:**

This paper contributes to LLM safety by investigates jailbreaking attacks. The study combines fine-tuning and accuracy-selective methods to enhances prompt effectiveness, broadening the range of attack techniques. Technique details, attack framework and evaluation metrics are all clearly explained in the proposal, make readers easy to follow and realize. I hope and have no doubt this project will achieve great results.

---

### Official Review · ~Jiuyang_Zhou1 · 2024-11-12
**Good direction and sufficient work**

**Rating:** 10
**Confidence:** 4

**Review:**

The paper focuses on the research of the hot and crucial topic of jailbreak attacks on large language models (LLMs). It aims to explore the performance of LLMs when facing such attacks and attempts to propose effective attack strategies by participating in relevant tracks of competitions and applying multiple methods. Overall, the research has strong practical significance and research value, providing useful ideas and exploration directions for further understanding and dealing with the security vulnerabilities of LLMs. The author has conducted sufficient research on the problem and investigated a lot of related work.

---

### Official Review · ~Isak_Tønnesen1 · 2024-11-12
**Clear and structured proposal.**

**Rating:** 9
**Confidence:** 4

**Review:**

This proposal addresses a crucial and timely topic of LLM security by investigating jailbreak attack methods. The methodology is well-structured, utilizing both white-box and black-box approaches, along with novel fusion methods combining fine-tuning and accuracy-selective techniques. The authors demonstrate thorough understanding of current attack strategies and provide clear evaluation metrics through the HEX-Phi scoring system. While the approach is comprehensive, incorporating more modern architectures like transformer-based models could strengthen the proposal. Overall, the practical significance for improving LLM safety and the clear technical framework make this a strong contribution to the field.

---

### Official Review · ~Maanping_Shao1 · 2024-11-12

**Rating:** 9
**Confidence:** 3

**Review:**

The proposal offers a timely exploration into the vulnerabilities of large language models (LLMs) concerning jailbreak attacks. By focusing on the CLAS 2024 jailbreak competition, the team aims to enhance LLM defenses by testing advanced attack techniques across a broad model set, including Llama-3 8B and other notable models. The methodology is well-structured, with clear definitions and a dual approach involving fine-tuning and accuracy selection for prompt injection techniques. This project is promising, with potential to yield valuable insights into improving LLM safety measures. However, additional clarity on anticipated challenges and limitations would strengthen the proposal's rigor.